# Steps to Facilitate the Use of Clinical Gait Analysis in Stroke Patients: The Validation of a Single 2D RGB Smartphone Video-Based System for Gait Analysis

**DOI:** 10.3390/s24237819

**Published:** 2024-12-06

**Authors:** Philipp Barzyk, Alina-Sophie Boden, Justin Howaldt, Jana Stürner, Philip Zimmermann, Daniel Seebacher, Joachim Liepert, Manuel Stein, Markus Gruber, Michael Schwenk

**Affiliations:** 1Human Performance Research Centre, Department of Sport Science, University of Konstanz, 78464 Konstanz, Germany; philipp.barzyk@uni-konstanz.de (P.B.); alina.boden@uni-konstanz.de (A.-S.B.); m.gruber@uni-konstanz.de (M.G.); 2Lurija Institute and Department of Neurological Rehabilitation, 78476 Allensbach, Germany; j.stuerner@kliniken-schmieder.de (J.S.); j.liepert@kliniken-schmieder.de (J.L.); 3Subsequent GmbH, 78467 Konstanz, Germany; philip.zimmermann@subsequent.ai (P.Z.); daniel.seebacher@subsequent.ai (D.S.); manuel.stein@subsequent.ai (M.S.)

**Keywords:** markerless motion capture, gait analysis, stroke, joint kinematics, RGB camera, human movement analysis

## Abstract

Clinical gait analysis plays a central role in the rehabilitation of stroke patients. However, practical and technical challenges limit their use in clinical settings. This study aimed to validate SMARTGAIT, a deep learning-based gait analysis system that addresses these limitations. Eight stroke patients took part in the study at the Human Performance Research Centre of the University of Konstanz. Gait measurements were taken using both the marker-based Vicon motion capture system and the single-smartphone-based SMARTGAIT system. We evaluated the agreement for knee, hip, and ankle joint angle kinematics in the frontal and sagittal plane and spatiotemporal gait parameters between the two systems. The results mostly demonstrated high levels of agreement between the two systems, with Pearson correlations of ≥0.79 for all lower body angle kinematics in the sagittal plane and correlations of ≥0.71 in the frontal plane. RMSE values were ≤4.6°. The intraclass correlation coefficients for all derived gait parameters showed good to excellent levels of agreement. SMARTGAIT is a promising tool for gait analysis in stroke, particularly for quantifying gait characteristics in the sagittal plane, which is very relevant for clinical gait analysis. However, further analyses are required to validate the use of SMARTGAIT in larger samples and its transferability to different types of pathological gait. In conclusion, a single smartphone recording (monocular 2D RGB camera) could make gait analysis more accessible in clinical settings, potentially simplifying the process and making it more feasible for therapists and doctors to use in their day-to-day practice.

## 1. Introduction

Stroke is the second leading cause of death and disability worldwide, with estimated costs exceeding USD 721 billion [1]. The rising occurrence of this condition highlights the need for an effective treatment strategy to enhance the quality of life for those affected and alleviate the burden on the healthcare system. The capacity to walk independently is a crucial goal for individuals who have experienced a stroke. However, in numerous instances, asymmetrical gait patterns, which are characterized by a reduction in walking speed, an increase in stride width, and a prolonged double support phase, persist in these individuals even after a considerable period of time [2]. Gait analysis systems can aid in precise and efficient diagnostics, therapy, and rehabilitation. Both lower extremity joint kinematics and spatiotemporal gait parameters are central to the assessment of the gait pattern in stroke patients and the quantification of disease status [3]. The use of instrumented gait analysis is the gold-standard method with which to examine gait deficits and direct subsequent rehabilitation and treatment options [4]. However, despite the availability of various systems, practical and technical hurdles such as complexity, costliness, and the effort required for installation and use have made it difficult for therapists and doctors to use them in standardized treatment [5,6,7,8]. Finally, the ability to prepare data correctly, interpret them accurately and identify errors is essential. However, in the majority of cases this done by clinical staff without specific training [6,7].

Several deep learning techniques have recently emerged as promising tools for efficiently processing instrumented gait data. They have the potential to address some of the existing problems with current gait analysis methods. A number of studies have explored the use of AI in this area [9,10,11,12,13], while others have highlighted the limitations of current methods [14,15,16]. To improve the feasibility of measurement, particularly in clinical settings, markerless gait analysis using low-cost equipment such as (smartphone) RGB cameras (capturing red, green, and blue wavelengths) is on the rise. However, only a few studies have applied their models to the analysis of pathological gait. Horsak et al. validated a markerless motion capture system (OpenCap, Stanford University Stanford, CA, United States) that uses two smartphone cameras and reported a root-mean-square error (RMSE) of ≤10.2° for lower extremity joint kinematics during impaired walking, suggesting the inaccurate estimation of the joint kinematics in cases of pathological gait [17]. As healthy individuals imitated gait disorders, the results of Horsak et al.’s study cannot be transferred to the clinical setting. Lonini et al. compared DeepLabCut (The Mathis Lab of Adaptive Intelligence, EPFL, Lausanne, Switzerland) as an open-source framework markerless pose estimation system against the established GAITRite^®^ system (CIR systems, West Conshohocken, PA, USA) in stroke patients [18]. They showed relative errors of 0.7 ± 18% for walking speed and 0.3 ± 5.9% for cadence, highlighting the potential of single-camera markerless pose estimation. However, they were not able to estimate spatial gait parameters with only one camera, as they could not obtain or calculate 3D models from 2D video sequences. Other studies on the application of markerless gait analysis to stroke patients also used two cameras [19] or multiple cameras [20], which makes the simple transfer of this technology to the clinic more difficult. The possibility of analyzing gait in a clinical setting using a single RGB camera (e.g., a smartphone) could make it much easier to use gait analysis in therapy and many patients could benefit from this.

In this study, we present SMARTGAIT, a new approach that uses deep learning algorithms to estimate individuals’ 3D joint coordinates from a single 2D RGB camera [21]. The main advantage of SMARTGAIT compared to other (markerless) systems is the ability to perform gait analysis using only a single handheld smartphone camera, without the need for additional hardware in the form of markers or sensors, statically positioned cameras, or complex calibration procedures. The SMARTGAIT trajectory reconstruction technique is based on a multi-stage convolutional neural network that estimates the 3D joint coordinates. The objective of this study was to compare the SMARTGAIT motion analysis technique against a gold-standard VICON system. The analysis was conducted on hip, knee, and ankle joint angular trajectories in the sagittal and frontal planes during overground walking, and used spatiotemporal gait parameters in stroke patients presenting pathological gait patterns.

## 2. Materials and Methods

### 2.1. Participants

Eight stroke patients (of which two were women) aged between 36 and 79 years (M = 59.4, ± 13.7 years) took part in this study at the Human Performance Research Centre (HPRC) of the University of Konstanz. The study’s protocol adhered to the Declaration of Helsinki for human experimentation and the ethical standards of the University of Konstanz. Each participant provided written informed consent before participating. Participant information and the severity of the stroke and impairment are listed in Table 1 and Table 2.

### 2.2. Study Design

Gait measurements were taken using both the marker-based Vicon reference system and the SMARTGAIT system (Figure 1). The camera was manually operated in conjunction with the movement of the participant. As precise positioning is not necessary, it merely requires the capture of the subject in the full scope. Ten gait trials were conducted on an eight-meter straight stretch. Participants walked barefoot if possible due to the severity of their condition; more detailed information on patient aids is listed in Table 1. They were instructed to walk at an “everyday, normal, and safe pace”. If necessary, a physiotherapist walked next to the participants for safety reasons, walking on the participant’s side furthest away from the SMARTGAIT cameras. Forty-three reflective markers (14 mm) were placed on anatomical landmarks according to the Plug-in Gait Full Body Model (Vicon Motion Systems, Oxford Metrics Group Ltd., Oxford, UK). Vicon markers were placed on top of an orthosis if needed. Only lower extremity angles were analyzed. To synchronize the systems, the timestamps of the foot contact made during the first gait cycle within the measurement area for both systems were determined. A least-squares fit was then used to determine the temporal offset. Additionally, the FMA-LE, FAC, and 10 m walk tests were completed on the same day or within 16 days before the gait measurement in the laboratory.

### 2.3. Material and Data Acquisition

#### 2.3.1. Vicon Motion Capture

The Vicon 3D motion capture system (Oxford Metrics, Oxford, UK) was used, comprising 12 Vicon-T40S infrared cameras positioned on the ceiling around the room with a sampling rate of 100 Hz and Vicon Nexus software (Version 2.12, Oxford Metrics, Oxford, UK). The system was calibrated before use and small 14 mm reflectors were utilized. Kinematics, joint angles, and gait parameters were calculated by the system based on the 3D coordinates of reflective markers placed at specific anatomical landmarks on each participant.

#### 2.3.2. SMARTGAIT

Similar to the Vicon Software, SMARTGAIT extracts 24 anatomical keypoint skeletal structures from the smartphone videos to estimate joint kinematics. The videos were recorded at 60 frames per second using a single Google Pixel 6a smartphone. The novel system analyzes gait and lower extremity kinematics from those videos in two stages. The process begins with a deep learning-based bounding box detector that identifies individual persons in the input video frames and generates initial bounding box estimates. From these detections, rectangular crops are extracted to isolate the regions of interest of all persons analyzed. These rectangular crops are then resized to a fixed target size. The second stage employs another deep learning model to predict key body points (24 in total) for each person in 3D relative to the camera [21]. For each person detected, a convolutional neural network (CNN)-based model is employed to predict a 3D heatmap volume that represents potential keypoint locations within a 3D space. This 3D heatmap volume encapsulates spatial information about each keypoint’s likelihood, enabling the model to make initial estimations about the positions of body parts. The use of 3D heatmaps provides a dense and informative representation that aids in distinguishing between overlapping joints and improves depth estimation accuracy. From the predicted 3D keypoint positions, we constructed a skeleton graph that captured the structural relationships between joints for each individual. This graph-based representation models the human body’s kinematic chain, facilitating downstream processing steps by maintaining the anatomical coherence of the pose. To refine the initial 3D positions and orientations of the skeleton, we leveraged learned anatomical priors that encode human body constraints and typical movement patterns. These priors help to adjust the estimated pose so that it aligns with feasible human postures, reducing noise and inaccuracies that may arise from ambiguous or partially occluded body parts. The resulting 3D keypoints are assembled into a complete 3D skeleton model for each frame of the video, resulting in 3D spatiotemporal skeletal movement data. The model was trained on a dataset containing around 1 million images of people performing everyday activities in a large variety of environments. Each image contains ground truth skeleton annotations, consisting of at least 24 keypoints for each person. This model is robust to challenges like perspective changes, occlusions, and clothing variations thanks to its use of statistical information about human movement. The resulting 3D skeletal movement data of the participants serve as the basis for further processing and subsequent gait analysis. Next, the data are normalized temporally and spatially in order to be invariant to differences in variables such as frame rate, body size, etc. Afterwards, these normalized data are used to perform gait cycle detection using a second deep learning model. The detected gait cycles are then used as the basis for assessing the gait parameters of the participants. For example, the step time is determined on the basis of two consecutive heel strike events of the opposite limbs, or in our case more generally based on two consecutive foot strike events, to enable the collection of data from participants suffering from foot drop paresis or other limitations.

#### 2.3.3. Statistical Analysis

The agreement between the SMARTGAIT and Vicon systems for measuring angular trajectories in the sagittal plane (i.e., hip flexion/extension, knee flexion/extension, plantar- and dorsiflexion) and frontal plane (i.e., hip abduction/adduction, knee varus/valgus), was quantified using Pearson correlations (r), the root-mean-square error (RMSE), the mean average error (MAE), and the absolute error in the maximal and minimal joint angles. Pearson’s r values ranging from 0.4 to 0.8 are considered to indicate moderate correlation, while values above 0.8 indicate strong correlation. Based on previous research on markerless motion [17,18], we expected a strong correlation between SMARTGAIT and Vicon. A priori power analysis with G*Power (Version 3.1.9.7) indicated that, with an expected correlation of 0.8, a significance level of 0.05, and a power of 0.80, a sample size of 8 is required.

Additionally, statistical parametric mapping (SPM) was used to determine significant agreement between the time-continuous measurements of the two systems using a *t*-test metric (*p* < 0.05). It is imperative to consider the results of the SPM in conjunction with the other metrics mentioned above given that significant outcomes may manifest with a minimal mean difference, but also due to a considerable standard deviation. To assess the agreement of the spatiotemporal gait parameters (step length, stride length, speed, cadence, stride time, and step time), we calculated intraclass correlation coefficients (ICCs) and the MAEs. ICC values between 0.75 and 0.9 indicate good agreement and values above 0.9 indicate excellent agreement [22]. RMSE values ≤ 2° were defined as low and RMSE values between 2 and 5 were defined as acceptable but requiring consideration in data interpretation after clinical gait analysis [23].

## 3. Results

### 3.1. Pearson Correlation

Correlations between Vicon and SMARTGAIT in the sagittal plane were 0.79 for all angle kinematics considered (Table 3). The maximum RMSE and MAE values were 4.6° and 3.2° for plantarflexion and dorsiflexion, respectively. In the frontal plane, we observed moderate to high correlations of 0.71, and the maximum RMSE and MAE values were 4.2°/3.0° for hip abduction and adduction. The analysis of the SPM revealed significant correlations between Vicon and SMARTGAIT for angular progressions in all planes (Table 3). Figure 2 illustrates the angular progression for the mean hip, knee, and ankle angles in the sagittal plane, as well as the mean hip and knee angles in the frontal plane, for a single participant across ten walks. For this example/participant, there is good agreement for both systems in the sagittal plane (Figure 2a,b), except for plantar/dorsal flexion (Figure 2c). In the frontal plane, there are greater deviations between the two systems for defined gait phases (Figure 2d,e). The figures for the remaining participants can be found in their original format in the Appendix A.

### 3.2. Intraclass Correlation

The ICCs (1,1) show good to excellent reliability for all gait parameters (Table 4). The mean speed and cadence values were 0.6 ± 0.4 m/s and 75.9 ± 24.7 steps/min for Vicon and 0.6 ± 0.4 m/s and 74.1 ± 26.4 steps/min for SMARTGAIT. The average stride and step times were 1.8 ± 0.7 s and 1.0 ± 0.5 s for Vicon and 1.9 ± 0.8 s and 0.9 ± 0.4 s for SMARTGAIT. In terms of spatial parameters, stride length and step length were 0.8 ± 0.3 m and 0.4 ± 0.1 m for Vicon and 0.8 ± 0.3 m and 0.4 ± 0.1 m for SMARTGAIT, respectively. The mean values for all gait parameters are presented in the Appendix B, along with the mean absolute error (MAE) values.

## 4. Discussion

The present study compared lower extremity kinematics and gait parameters in stroke participants using SMARTGAIT, a novel deep learning-based system for gait analysis, against a gold-standard Vicon system. The results demonstrated moderate to high levels of agreement between the two systems, with Pearson correlations of 0.79 for all lower body angle kinematics in the sagittal plane and 0.71 in the frontal plane. The correlations found between both systems may suggest that SMARTGAIT has the potential to measure lower extremity angle trajectories during gait cycles with an accuracy that is sufficient for clinical gait analysis, particularly in the sagittal plane. In addition to joint kinematics, the study also evaluated the agreement of spatiotemporal gait parameters between the SMARTGAIT and Vicon systems. The results showed excellent agreement for all parameters (ICCs 0.96), except for step length (ICC = 0.78). These findings suggest that SMARTGAIT provides moderate to excellent accuracy when estimating joint kinematics and spatiotemporal gait parameters, even from videos taken on a single 2D RGB smartphone camera.

The use of deep learning algorithms in gait analysis has been explored in previous studies [9,10,11,12,13], but the application of these techniques to clinical settings has been limited due to practical and technical challenges. The current study addresses these limitations by demonstrating the proof of concept of using a single RGB camera for gait analysis.

### 4.1. Kinematics

The majority of motion during normal walking occurs in the sagittal plane. This provides useful information, for example, for quantifying the status of neurological disease [3]. McGinley et al. proposed that, in the majority of clinical scenarios, errors of 2° or less are likely to be deemed acceptable, as such errors are likely to be too minor to necessitate explicit consideration during data interpretation [23]. Errors between 2° and 5° are also likely to be regarded as reasonable, although they may require consideration during data interpretation. The errors observed in our study fell within the 2° to 5° range, indicating that SMARTGAIT may have clinical value for gait analysis. However, caution is required when interpreting values. This is particularly relevant to the plantar dorsi flexion angle (RMSE 4.6°), which has a relatively small range of motion (ROM), thereby increasing the impact of measurement errors.

The RMSE values were found to be in the range of 2° to 5°, which is comparable to the results of Horsak et al., who demonstrated RMSE values for hip abduction between 3.7° and 4.8° in the analysis of three pathological gait patterns [17]. Although the results demonstrate good values throughout the gait cycle, the results cannot be readily generalized. As illustrated in Figure 2d,e, the novel system encounters significant challenges when calculating movements, such as abduction and adduction or varus and valgus of the knee. This can lead to complications when analyzing individual sections of the gait cycle. 

Horsak et al. employed two smartphone cameras and OpenCap markerless gait analysis [17]. The healthy participants simulated the three pathological gait patterns of crouch, circumduction and equinus. In general, they observed greater RMSE values for hip flexion (ranging from 5.5° to 7.6°), knee flexion (5.9° to 8.5°), and ankle angle (6.1° to 7.9°) compared to the findings in our study. This suggests that the pose estimation algorithms embedded into SMARTGAIT result in higher correlation and lower error rates during marker-based motion capture than previous approaches, despite the use of fewer cameras (i.e., one vs. two cameras). One can hypothesize that the “end-to-end” modeling of the 3D reconstruction within the deep learning model of SMARTGAIT is more robust than the retrospective reconstruction of multiple 2D views of OpenCap.

Moreover, the findings of Horsak et al. demonstrate particularly high RMSE values for the joint most affected by the pathological gait pattern (crouch = knee flexion; equinus = ankle flexion) [17]. It is evident that deep learning algorithms encounter challenges in accurately identifying aberrant gait patterns, as these are not as extensively represented in the datasets upon which the systems are based [14]. Nevertheless, this outcome also suggests the potential for these methods to yield enhanced outcomes with the inclusion of further pathological gait data.

### 4.2. SPM Analysis

The average SPM values did not exceed *p* ≤ 0.05 for any angle trajectory, indicating agreement between SMARTGAIT and the Vicon system. Few SPM values in individual patients were above the threshold for specific angles (see Appendix B). This shows that deviations can occur at specific phases of the gait cycle in individual cases, and these are more significant. These deviations were not observed more frequently at a specific angle or in a specific patient. A potential cause of these outliers may be that patients perform special movements due to their stroke, which the deep learning models only recognize to a limited extent. However, these outliers may be fully compensated for by including additional training data of patients with pathological gait patterns, as discussed above.

### 4.3. Comparison of Patients with and Without Walking Aid

A comparison of the data in Table 3 revealed that patients with a walking aid exhibited a greater incidence of measurement errors (RMSE range: 2.8–6.6°) than patients without a walking aid (RMSE range: 2.6–5.2°). In patients who were not using a walking aid, all RMSE values were below the 5° threshold deemed suitable for clinical applications (with the exception of one patient who displayed hip abduction/adduction: RMSE: 5.2°). Conversely, 35% of the RMSE values obtained with a walking aid were above the threshold value of 5°, with the majority of these values being above the threshold for plantar/dorsiflexion and hip abduction/adduction. The increased occurrence of elevated RMSE values in these patients may be attributed to a number of factors. The use of a walker or the presence of a therapist, particularly on the side facing the camera, can result in increased masking and, on occasion, a lack of contrast between the walker/therapist and the legs or shoes and the background. These factors, in conjunction with the observation that all patients with gait abnormalities exhibited the most aberrant and deviant gait patterns of all patients studied, may account for the increase in RSME scores observed.

### 4.4. Spatiotemporal Gait Parameters

We observed excellent ICC values for all gait parameters, except for step length, demonstrating the potential of SMARTGAIT for measuring spatiotemporal features during walking. Nevertheless, our results also show that there is potential for improvement in the assessment of specific spatial gait parameters. While the assessment of the stride length works well, the accuracy of the step length estimation is limited. This probably relates to the significant asymmetries in the gait pattern of stroke patients, which are challenging for the SMARTGAIT system to recognize.

Lonini et al. employed DeepLabCut, an open-source framework for markerless pose estimation, to track five body keypoints (hip, knee, ankle, heel, and toe) in stroke patients [18]. As a reference, the temporal gait parameters were then compared to those obtained using the GAITRite system. In contrast to SMARTGAIT, the pre-trained deep learning model was further fine-tuned via the manual annotation of the positions of five landmarks on the leg and foot in two frames for each video. Furthermore, videos from both sides were included in the dataset. The systems were thus able to detect the parameters regardless of the presence of assistive devices and other people in the scene.

The MAE and SD values of our results (0.01 ± 0.03 m/s) were comparable to the results reported by Lonini and colleagues (−0.02 ± 0.11), whereas the MAE values of cadence was found to be significantly higher in the present study (1.80 ± 4.27 vs. −0.25 ± 3.88) Notwithstanding the tendency of both deep learning algorithms to underestimate gait parameters in comparison to the reference systems, according to Lonini et al., manual input of marker positions is necessary to refine their outcomes [18]. Consequently, SMARTGAIT could offer advantages in terms of ease of use, which would be beneficial for practical application. Furthermore, Lonini et al. were unable to obtain or calculate three-dimensional models from their two-dimensional video sequences, which precluded the analysis of spatial gait parameters [18].

In future, we are considering several promising ways to further improve or adapt the methods used in SMARTGAIT. For example, a more explicit temporal modeling of the movement characteristics could be added to the pose estimation model in order to better capture fine-grained temporal movement patterns, especially in situations with partial occlusions. Furthermore, task-specific fine-tuning using the acquired motion capture data could be performed to develop a model which is more specialized in gait movements. Such specialized models could then be used to capture additional types of gait and movement parameters beyond the scope of SMARTGAIT. 

## 5. Conclusions

The findings of this study have several implications for clinical practice. Firstly, the use of a single 2D RGB camera for gait analysis could make gait analysis more accessible in clinical settings. This could potentially simplify the process and make it more feasible for therapists and doctors to use in their practice. Secondly, traditional gait analysis systems like Vicon are often expensive and require specialized equipment and trained personnel. In contrast, a system like SMARTGAIT that uses simple smartphone videos could be a more cost-effective solution. Thirdly, the ability to accurately estimate joint kinematics and spatiotemporal gait parameters could aid therapists and doctors in diagnosing and treating conditions like stroke-induced motor deficits. It could also be used to monitor participant progress during rehabilitation. Lastly, the success of the SMARTGAIT system could encourage further research into the application of deep learning algorithms in gait analysis. This could lead to the development of even more accurate and efficient systems in the future.

Nevertheless, expertise in data collection and analysis remains of significant importance. Despite the ease of use and efficiency of the novel system, the generation of good results is contingent upon its correct application and the interpretation of its data. Furthermore, it is important to note that while the correlations were predominantly high, there were still some discrepancies between the SMARTGAIT and Vicon systems. These issues may be due to the inherent limitations of estimating 3D joint coordinates from 2D images. Future research could focus on improving the accuracy of these estimations, perhaps by incorporating additional data or refining the deep learning algorithms used. Nonetheless, the lower limb kinematics errors, measured in degrees, were below the clinically desirable threshold of five degrees for all angles [23].

The results of this study suggest that SMARTGAIT is a promising tool for gait analysis in stroke. Further analyses are required to validate SMARTGAIT in larger samples and evaluate its transferability to different forms of pathological gait.

## Figures and Tables

**Figure 1 sensors-24-07819-f001:**
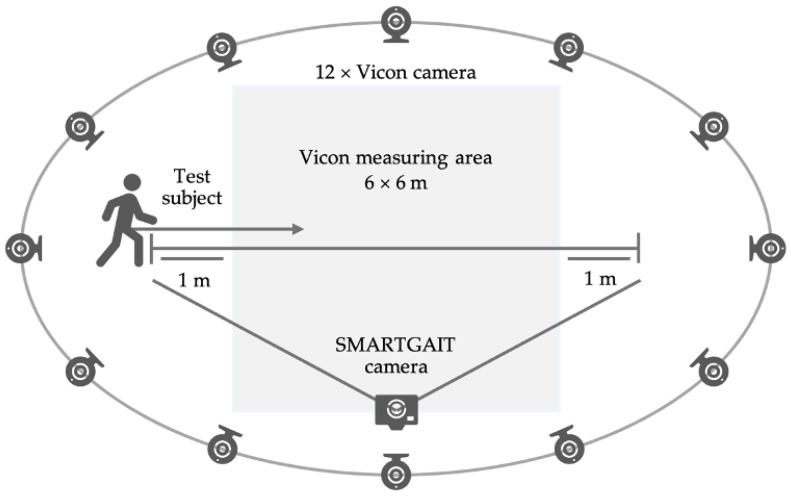
The image depicts the study design setup, with its 12 Vicon cameras (circular camera icons), and the SMARTGAIT recording position (square camera icon) within the subject’s sagittal plane, along with the subject’s walking distance. The data were collected within a 6 × 6 Vicon area, with a 1 m run-out zone in front of and behind it.

**Figure 2 sensors-24-07819-f002:**
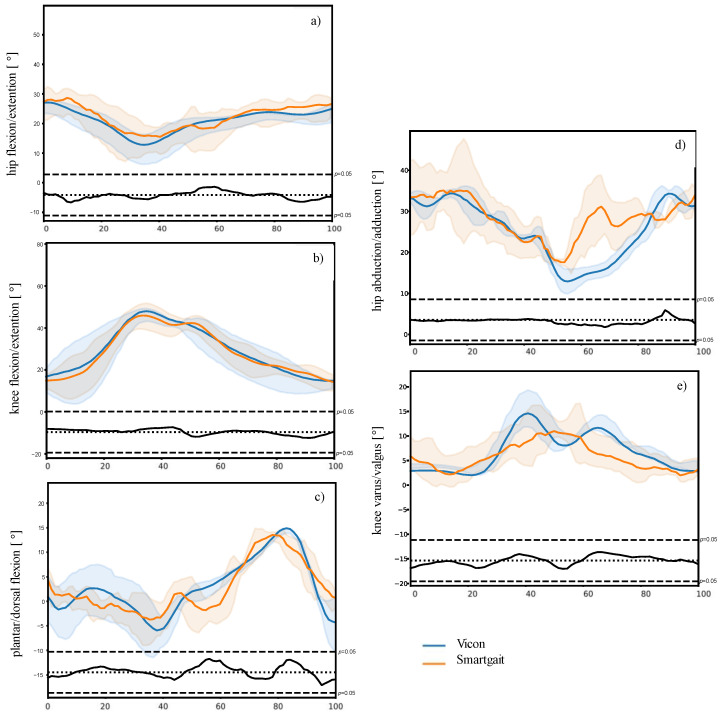
Mean (**a**) hip, (**b**) knee, and (**c**) ankle angles in the sagittal plane, and mean (**d**) hip, and (**e**) knee angles in the frontal plane, exemplarily, for participant P01 across ten walks. The shaded area around the curve represents the variance within the data. Zero percent gait cycle indicates initial foot contact, and 100 percent indicates the foot is off. Zero degrees means full hip and knee extension and neutral ankle position. The gray bar and black line at the bottom of each graph indicates significance (from SPM analysis) for the corresponding t-test statistics in relation to a critical threshold (black dashed lines; *p* = 0.05).

**Table 1 sensors-24-07819-t001:** Participant information.

Participant	Age [Years]	Sex	Weight [kg]	Height [mm]	Walking Aid
P01	79	m	71	171	No
P02	44	m	78	170	No
P03	36	m	76	171	No
P04	62	m	79	170	No
P05	60	m	80	163	Walking stick and foot lift orthosis
P06	62	f	60	166	Quad stick and foot lift orthosis
P07	61	f	65	160	Quad stick and foot lift orthosis
P08	71	m	93	174	Walking stick and foot lift orthosis

**Table 2 sensors-24-07819-t002:** The severity of the stroke and impairment.

Participant	Time Since Stroke [Months]	Paretic Bodyside	Type of Stroke	Fugl–Meyer Lower Extremities [FMA-LE]	Functional Ambulation Categories [FAC]	10-m Walk Test [10MWT, s]
P01	51	right	Ischemic	21	5	n.r.
P02	1	left	Hemorrhagic	26	4	15.6
P03	4	right	Hemorrhagic	31	5	7.15
P04	130	left	Ischemic	20	5	n.r.
P05	4	right	Hemorrhagic	13	2	41.2
P06	4	left	Ischemic	18	3	35.7
P07	4	left	Hemorrhagic	19	3	28.55
P08	3	left	Hemorrhagic	17	3	20.85

The FMA-LE scale is an index used to evaluate sensorimotor impairment in individuals who have suffered a stroke (scores of 0–17 = severe disability; scores of 16–22 = marked disability; scores of 23–28 = moderate disability; scores of 29–33 = mild disability; score of 34 = normal function). The FAC is a six-point functional walking test that assesses a participant’s ability to walk and determines the level of human support needed (ranging from 0 (non-functional) to 5 (independent)). The level of mobility is determined by the need for assistance. A score of 4 or higher indicates the ability to walk independently. The 10MWT assesses locomotor capacity by calculating the time it takes a person to walk a set distance of 10 m (n.r. = not reported).

**Table 3 sensors-24-07819-t003:** Statistical results for the agreement between SMARTGAIT and the Vicon system.

Plane	Movement	Pearson’s r	RMSE [°]	MAE [°]	Max_err [°]	Min_err [°]	SPM [*p*-Value]
Sagittal	Hip flex./ext.	0.95	3.5	2.7	2.6	2.5	0.011
	Knee flex./ext.	0.94	3.7	2.8	2.4	2.9	0.009
	Plantar/dorsi flex.	0.79	4.6	3.2	2.1	4.5	0.012
Frontal	Hip abd./add.	0.75	4.2	3.0	2.3	2.2	0.009
	Knee var./val.	0.71	3.9	2.9	3.7	2.8	0.008

Pearson correlations (r), root-mean-square error (RMSE), mean absolute error (MAE), and angular difference in the maximum (Max_err) and minimum (Min_err) angular position joint angles, as well as statistical parametric mapping (SPM) results.

**Table 4 sensors-24-07819-t004:** Intraclass correlation coefficients for agreement between Vicon and SMARTGAIT.

Parameters	ICC (1,1) Point Estimate	Lower 95% CI	Upper 95% CI
Cadence (steps/min)	0.987	0.971	0.994
Stride time (s)	0.982	0.960	0.992
Step time (s)	0.963	0.918	0.984
Speed (m/s)	0.997	0.994	0.999
Stride length (m)	0.985	0.967	0.994
Step length (m)	0.781	0.562	0.899

Values greater than 0.90 indicate excellent agreement.

## Data Availability

The data presented in this study are available on request from the corresponding author due to legal reasons.

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
