# Peer review of "Steps to Facilitate the Use of Clinical Gait Analysis in Stroke Patients: The Validation of a Single 2D RGB Smartphone Video-Based System for Gait Analysis"

_sensors, 2024, doi:10.3390/s24237819_

Round 1

Reviewer 1 Report

Comments and Suggestions for Authors

The authors peent a study designed to explore the validity and reliability of a marker less smartphone-based gait analysis system compared to a traditional video gait analysis system. The smartphone-based system employed a deep learning algorithm to determine joint coordinates. The use of smartphone-based motion tracking has become more common due to the relative ease of use and lower costs. The current paper does help address the concerns regarding the validity and reliability of the smartphone-based systems in a clinical population (patient recovering from stroke). The application of the smartphone-based motion tracking will help aid in the research and treatment of this clinical population. The authors present a traditional validity and reliability investigation measuring the joint kinematics with both systems at the same time. The variability in the joint kinematics in this clinical population could present challenges to the reliability of these measures. The authors present the ICC and the analysis of the root mean square that are useful to the researcher. The calculation of the minimal detectable change and is missing and might be more useful to the clinician wishing to use the smartphone-based system. The authors present the tables and figures to help present their results. A few specific concerns are list below.

Line 162 missing reference.

Lines 162-164 suggest adding more information regarding the power analysis.

Line 170 inconsistent reference style the rest of the paper uses a numbered style line 170 has an author and date style.

Line 175 a superscript 3 appears after the word were

Line 178 same after word of

Line 180 I think table 2 should be table 3

Line 212, 213 219 superscript 3 again

Author Response

Dear reviewer, Thank you for your valuable comments and remarks, we have included specific responses to your comments in red in the attached document. We believe that your comments have significantly 

Reviewer 2 Report

Comments and Suggestions for Authors

Paper Title:

Steps to facilitate the use of clinical gait analysis in stroke patients: Validation of a single 2D RGB Smartphone Video-Based System for Gait Analysis

This study is an intriguing exploration of the effectiveness of the SMARTGAIT system, which utilizes a single 2D RGB camera for gait analysis in stroke patients, compared with the conventional, costly, and complex Vicon motion capture system. Nevertheless, there are considerable concerns, particularly with regard to the methodology.

Major Comments:

The manuscript is rendered quite confusing by discrepancies in the numbering of tables and their references within the text. Please verify these details prior to resubmission.

Figure 1-d illustrates a notable discrepancy between the Vicon and SMARTGAIT data during the gait cycle at 60-80%. However, the SPM{t} data indicates minimal fluctuation, suggesting no discernible difference. In contrast, in Figure1-a, where the two datasets appear more aligned, the SPM{t} variation appears to be larger than in Figure1-d. It is imperative that the threshold values of SPM{t} in each figure be specified and that the analysis be conducted correctly. Furthermore, the vertical axis for SPM{t} in all of Figure 1's sub-figures lacks clarity. In Figure 1-e, the data appears to be truncated at approximately 90%. A review of the figures, tables, data, and analysis methods is required. Furthermore, the limited number of trials and high degree of data variability may result in differences between Vicon and SMARTGAIT being less pronounced.

Minor Comments:

Page 4, line 133: The procedure for setting up SMARTGAIT is not readily apparent. It would be beneficial to include a visual representation of the setup method in the "2.3.2" section. The section on SMARTGAIT should be revise

Page 5, line 176: Is it possible that the correct reference is "Table 3" instead of "Table 2"? Please verify this information, as well as that on line 180.

Page 5, Table 3: on page 5 contains instances where “Min_err” exceeds “Max_err”. Please clarify whether this is an error.

Page 5, Table 3: Could you please clarify how the r-value was calculated? Does the SPM calculate the correlation coefficient, r? If this is the case, can you please clarify whether this is the mean value across the eight subjects or the value for subject P01?

Page 7, Table 3: It appears that an error has been made; should the table in question be labelled as Table 4 instead of Table 3? Additionally, line 197 on page 6 requires correction.

As previously stated, the interpretation of results was not entirely clear, making it challenging to assess the discussion. Consequently, I would appreciate the opportunity to review the discussion in light of the aforementioned clarifications. 

Author Response

(The authors gave the same response as above.)

Reviewer 3 Report

Comments and Suggestions for Authors

The focus of the paper is to validate a single 2D RGB smartphone video-based system for gait analysis. Since the technology utilized is already established, the primary contributions of the paper are the collection of real-world data and the provision of comparative discussions from multiple perspectives. Overall, there is considerable room for improvement, and the paper does not align well with the journal's scope; therefore, it is advised to consider submission to journals more focused on biomedical engineering unless additional technical content is included. Nonetheless, the reviewer offers the following suggestions for improvement that the authors should consider:

1. The tables are improperly formatted. They lack titles, and the numbering of the footnotes is not reflected within the tables.

2. The section 2.3.2 is repeated.

3. Technical details related to smartGait are insufficient, including aspects such as data processing, model architecture, hyperparameters, and the implementation of unmarked training, among others.

4. The methods for calculating evaluation metrics are not provided.

5. It is unclear which subject in the Appendix uses walking aids and which does not.

Author Response

(The authors gave the same response as above.)

Round 2

Reviewer 3 Report

Comments and Suggestions for Authors

The newly added Figure 1 is very helpful in enhancing the reader's understanding of the implementation of the work. However, the authors should take a more rigorous approach to the writing and revision of the paper. For instance, the references to Figure 1 in the text do not match, and Figure 2 is not adequately explained in the manuscript.

According to the added technical details about smartGait, the main contribution of the paper lies in the transfer and innovative application of existing algorithms. Have the authors considered any strategies to improve accuracy and make smartGait closer to the gold standard?

Author Response

Thank you very much for your additional remarks, we incorporated them in our newly revised manuscript:

1. We added the following for Fig.2, to give further information to the reader: "Figure 2 illustrates the angular progression for the mean hip, knee, and ankle angles in the sagittal plane, as well as the mean hip and knee angles in the frontal plane for a single participant across ten walks. For this example-participant, there is good agreement for both systems in the sagittal plane (Figure 2 a, b), except for plantar/dorsal flexion (Figure 2c). In the frontal plane, there are greater deviations between the two systems for defined gait phases (Figure 2d, e)."

2. We added a paragraph at the end of our discussion to go further into detail on how SMARTGAIT can be improved: "For the future, we are considering several promising ways to further improve or adapt the methods used in SMARTGAIT. For example, a more explicit temporal modeling of the movement characteristics could be added to the pose estimation model in order to better capture fine-grained temporal movement patterns, especially in situations with partial occlusions. Furthermore, task-specific fine-tuning using the acquired motion-capture data could be performed to develop a model which is more specialized to gait movements. Such specialized models could then be used to capture additional types of gait and movement parameters beyond the scope of SMARTGAIT."